# A Visual Analysis Method for Predicting Material Properties Based on Uncertainty

**Qikai Chu** [1,2] , **Lingli Zhang** [2,3,†] , **Zhouqiao He** [2,3] , **Yadong Wu** [2,3,*,†] and **Weihan Zhang** [2,3]

1  School of Automation and lnformation Engineering, Sichuan University of Science and Engineering, Yibin 644002, China
2  Sichuan Provincial Engineering Laboratory of Big Data Visual Analysis, Yibin 644002, China
3  School of Computer Science and Engineering, Sichuan University of Science and Engineering, Yibin 644002, China
*  Correspondence: wyd028@163.com
†  These authors contributed equally to this work.

**Abstract:** The traditional way of studying fluorinated materials by adjusting parameters throughout multiple trials can no longer meet the needs of the processing and analysis of multi-source, heterogeneous, and numerous complex data. Due to the high confidentiality of fluorinated materials' data, it is not convenient for the plant to trust the data to third party professionals for processing and analysis. Therefore, this paper introduces a visual analysis method for material performance prediction supporting model selection, MP²-method, which helps with researchers' independent selection and comparison of different levels of prediction models for different datasets and uses visual analysis to achieve performance prediction of fluorinated materials by adjusting control parameters. In addition, according to the Latin hypercube Markov chain (LHS-MC) model of uncertainty for visual analysis proposed in this paper, the uncertainty of the control-parameter data is reduced, and their prediction accuracy is improved. Finally, the usefulness and reliability of MP²-method are demonstrated through case studies and interviews with domain experts.

**Keywords:** machine learning; autonomous model selection; visual analytics; performance prediction; uncertainty analysis

## 1. Introduction

Fluorinated materials are compounds containing fluorine elements, among which, high-end fluorinated materials are widely used in the field of national defense [1], and their technological breakthroughs have become a common pursuit of countries around the world. As the research proceeds, the control parameters and performance data of various fluorinated materials are accumulating, the amount of data is increasing dramatically, and the access to and analysis of data are becoming increasingly difficult and inefficient when adjusting control parameters or designing new research routes, which prolongs the time from laboratory development to industrial production and limits the progress of fluorinated materials' development [2].

As for research on performance prediction for materials based on machine learning, previous studies have focused on the comparative study of different machine learning models [3–6], through the selection of model parameters [7], optimization of features [8], and comparison of accuracy, to select machine learning models that are most suitable for the corresponding material data in question. These studies have theoretically improved the efficiency of material product development and relatively reduced the number of manual experiments. However, these model-optimal comparison experiments are often only for the overall performance of a particular dataset. There are many uncertainties in the process of material product development. Previous studies cannot show the specific details of a

single factor in the product prediction process, which is one of the most critical factors in model selection for materials science researchers [9].

In this paper, we propose a visual analysis method for performance predictions of fluorinated materials, MP$^2$-method, which uses machine learning and deep learning to explore the relationships between fluorinated materials' control parameters and product performance to achieve product-performance prediction. Considering that parameters have many uncertainties, the Latin hypercube Markov chain (LHS-MC) uncertainty-visualization model proposed in this paper introduces the uncertainty visualization method [10], which helps users to visually identify the instability of control-parameter data and the uncertainty of product-performance prediction. In addition, MP$^2$-method changes the traditional situation of handing material development data to professionals for model prediction, facilitating the independent selection of suitable machine learning models for one's needs [11] and achieving accurate forecasting through fine adjustment of the parameters. According to our research, MP$^2$-method is the first interactive visual analysis system for fluorinated-material performance prediction and supports different types of machine learning models.

This work's significant contributions are summarized below:

- A new approach to fluorinated-material performance-prediction research is proposed, and an interactive visual analysis system is developed to support the user's independent selection of performance-prediction models according to their needs, to evaluate the merits of the models comprehensively and to assist in the development of fluorinated-material products.
- The Latin hypercube Markov chain (LHS-MC) visual uncertainty analysis model is proposed to assist users in the visual analysis of the performance predictions of fluorinated materials by calculating and transferring uncertainty between each population in the development process.
- An evaluation of performance prediction and uncertainty visualization methods, including case studies using actual data from fluorochemical production plants and interviews with experts in the chemical field.

## 2. Related Work

The research of our work can be divided into the following three parts: visual analysis of association mapping, performance prediction analysis, and visual analysis of uncertainty.

- Visual Analysis of Association Mapping.

Multi-attribute correlation visual analysis refers to the use of visualization methods to visually analyze the correlations between attributes and the influencing factors between them, and help users discover the implicit mapping relationships and pattern characteristics in the data. Traditional correlation visual analysis methods include: a scatter plot matrix, a parallel coordinate technique, an adjacency matrix [12], a node-link diagram [13], a chord diagram [14], a tree diagram [15], and other types. Different methods of association visual analysis apply to different scenarios. A scatter diagram matrix is used to analyze the association relationship between two attribute data as a group. Parallel coordinates are used to analyze the association between multiple attributes by connecting parallel axes, but the problem of "dimensional disaster" occurs when facing high-dimensional data, which makes the multi-dimensional attribute analysis unsatisfactory. Therefore, in this paper, we propose a multi-attribute-correlation, visual analysis-based decision method, which is based on the premise of node connection and optimizes the ordinary parallel coordinates to explore the correlations between the factors influencing multiple attributes of fluorine chemical products. It should help researchers to make quick decisions.

- Performance Prediction Analysis.

The field of materials development incorporating machine learning models to accelerate materials design and application has seen a great deal of research done by experts in the area. The study presented in [16] used a random forest model to predict yield strength and elongation, respectively, to improve extension and yield length. The study presented

in [17] used artificial neural networks to predict product performance and improve product-development efficiency. Another study [18] found that machine learning models' prediction performances varied depending on the dataset.

In this paper, we propose a visual analysis method for performance prediction, which helps users to make targeted selections of performance-prediction models and assists users in quickly filtering the machine learning models with the best prediction results.

- Visual Analysis of Uncertainty.

In the various stages of fluorinated materials' development, especially in product development, polymerization control, the polymerization process, and performance testing, there are many uncertainty problems in data processing, parameter measurement, and management, which affect the accuracy of product performance assessment and prediction. Uncertainty visualization has become a relatively popular research direction in the visualization field. Most of the previous authors have reduced data analysis errors and improved research accuracy by mining the uncertainty of analytical data [19], and visual graphics and interactive methods are the preferred approaches of researchers for visual uncertainty analysis [20], including circular tree diagrams [21–23] and many other visual analysis methods that are used to demonstrate the uncertainty in their development process visually.

In this paper, we propose the Latin hypercube Markov chain (LHS-MC) visual uncertainty analysis model for the study of fluorinated-material performance predictions. It uses uncertainty-visualization models [24] to visually reveal the uncertainty in the development data, provide the correct references for the decision analysis of the core aspects of the development process, and improve the agreement between the theoretical model of the development process and the actual experimental results.

## 3. Performance-Prediction Model's Design

### 3.1. Data Preprocessing

In this study, we systematically sorted and analyzed relevant information from domestic and foreign public databases, professional books, and encyclopedic resources, such as the China National Materials Science Data Sharing Network, Fluorosilicon Materials Network, MatWeb, Total Materia, Matmatch, fluoropolymer properties database, and Wikipedia. Using Mysql as the storage medium, we established a basic knowledge base for the development of fluorinated materials and provided efficient and reliable data support for the performance prediction and verification of fluorine materials.

After interviews with industry experts, as shown in Table 1, four subsets were established from the entire dataset, with each section including a training set and a test set—80% for the training set and 20% for the test set. For the first two subsets, the four characteristics of reaction temperature, PH regulator type and amount, stirring speed, and monomer type were used as influencing factors; and Menny viscosity and stretching length were used as predictive values. For the latter two subsets, we took four characteristics—reaction temperature, type of initiator system, reaction pressure, and type of reactor—as influencing factors, and Mennian viscosity and glass transition temperature as predictive values. The first and third subsets accounted for 50% of the entire dataset, and the second and fourth subsets accounted for 100% of the entire dataset.

### 3.2. Building a Performance Prediction Model

Based on previous studies, we propose a visual analysis method for performance prediction supporting user-autonomous model selection, and 10 machine learning models were established to be included in the method. We briefly introduce the research work on two models, the BP neural network and random forest, as examples. We built the models to predict the performances on four datasets and visualize and compare the prediction values, mean square error, and other process data of each trained model. Additionally, an uncertainty visualization and analysis model was built to reduce the effects of the

changes in external influences considering the momentary changes in temperature in the environment.

**Table 1.** Fluorinated-material dataset's division.

| Dataset | Percentage of Data Volume | Influencing Factors | Predictive Performance | Amount of Data | Data Dimension |
|---|---|---|---|---|---|
| Data Set-1 | 50% | Response temperature<br>PH regulator type and addition amount<br>Stirring speed<br>Monomer type | Menny viscosity<br>Stretching length | 4950 | 34 |
| Data Set-2 | 100% | Response temperature<br>PH regulator type and addition amount<br>Stirring speed<br>Monomer type | Menny viscosity<br>Stretching length | 10,230 | 34 |
| Data Set-3 | 50% | Response temperature<br>Types of elicitor systems<br>Reaction pressure<br>Reaction Kettle Types | Menny Viscosity<br>Glass transition temperature | 5020 | 34 |
| Data Set-4 | 100% | Response temperature<br>Types of elicitor systems<br>Reaction pressure<br>Reaction Kettle Types | Menny Viscosity<br>Glass transition temperature | 9897 | 34 |

### 3.2.1. Back-Propagation Neuron Network (BPNN) Model

A BP neural network (BPNN) consists of three parts: the input layer, the hidden layer, and the output layer. The number of nodes in the hidden layer is the most critical factor affecting the capacity of the BPNN. In this study, the sigmoid function, the S-shaped transfer function, was used as the gradient function for the network. Additionally, we used the empirical formula Equation (1) to calculate this training. When the number of neurons in the hidden layer was three, the model's effect was relatively better, so the model with three layers of neurons in the hidden layer was used in this study, as shown in Figure 1.

$$T(x) = \frac{1}{1 + \exp(-x)} \tag{1}$$

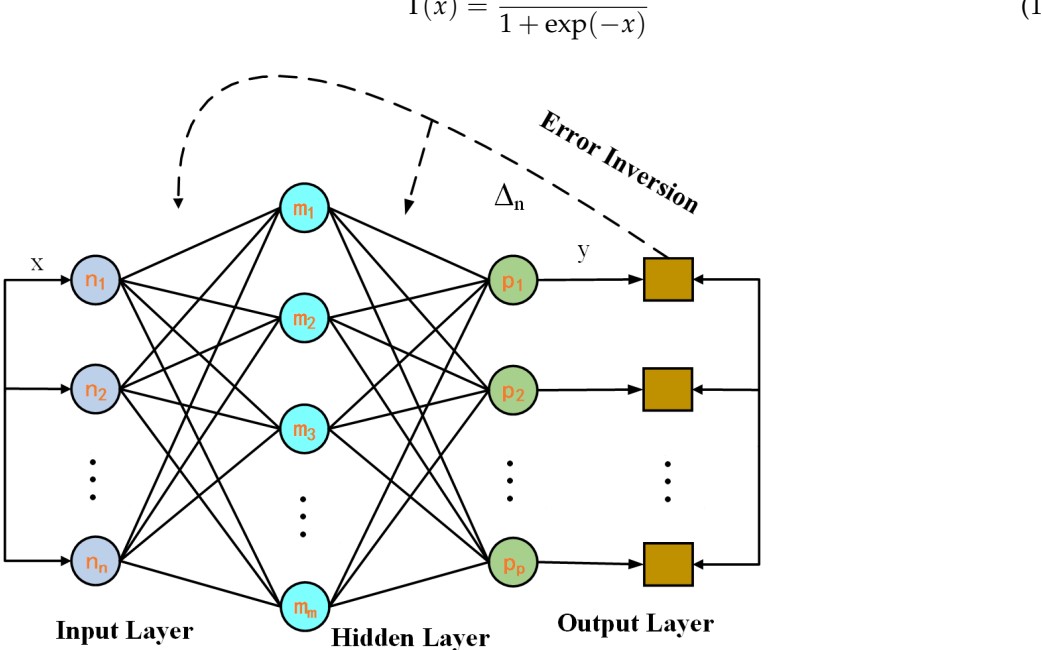

**Figure 1.** The input layer's neurons are represented by light purple, blue represents the implicit-layer neurons, and green represents the output-layer neurons.

The choice of the number of nodes in the hidden layer is crucial for the construction of the model, which determines the performance of the model. We took a relatively small number of nodes in the hidden layer under the premise of satisfying the accuracy as much

as possible and used the empirical formula (Equation (2)) to show that the model's effect is comparatively better when the number of neurons in the hidden layer is three. This was verified in the later case study through the implementation of the experiment.

$$M_{\text{hidden}} = \sqrt{M_{input} + M_{output}} + \gamma \qquad (2)$$

The number of neurons in the buried layer is denoted by $M_{\text{hidden}}$; $M_{input}$ and $M_{output}$ represent the numbers of input and output neurons, respectively; $\gamma$ represents a constant between 1 and 10. Therefore, In this research, a model (Figure 1) with three layers of neurons in the hidden layer was adopted.

Calculated by the BP neural network algorithm, shown as a flowchart in Figure 1, the hidden layer's mth neuron receives the input layer's input, as shown in Equation (3), and the output layer's path neuron receives the input as shown in Equation (4).

$$\beta_n = \sum\nolimits_{n=1}^{k} \omega_{nm} z_n \qquad (3)$$

$$\partial_p = \sum\nolimits_{m=1}^{n} \omega_{mp} z_m \qquad (4)$$

Orange indicates the error contained in the output layer, and the error is passed back to the implicit neuron through the inverse error function $E(x)$, as shown in Equation (5) below. The weights and thresholds are adjusted according to the returned errors, and the cycle is continuously iterated so that the function avenue is minimized.

$$E(x) = \frac{1}{2} \sum\nolimits_{i=1}^{p} (\alpha_i - \beta_i)^2 \qquad (5)$$

$\alpha_i$ is the desired output, and when we denote $b - a$ as e, the inverse error function $E(x)$ is expressed as shown in Equation (6). Then, we update the weight values as shown in Equations (7) and (8), where $\eta$ is the learning rate.

$$\omega_{nm} = \omega_{nm} + \eta \beta_n (1 - \beta_n) z_n \sum\nolimits_{n=1}^{k} \omega_{nm} e_i \qquad (6)$$

$$\omega_{mp} = \omega_{mp} + \eta \beta_n e_i \qquad (7)$$

Although the BPNN model has strong nonlinearity and generalization, it also has many disadvantages, such as low global search ability and slow convergence speed. Therefore, with the advantage of genetic algorithm, we find a relatively good search space for the BPNN model in the analytic space and then search for the optimal solution.

### 3.2.2. Random Forest (RF) Model

The random forest algorithm is essentially a combination of K samples that are automatically sampled and repeated in the original sample data to form a new sample. Then, N new features are randomly selected, and the best ones from these features are chosen as samples. Each sample is individually constructed into a decision tree, and finally, these decision trees are randomly voted on, as shown in Figure 2.

The random forest algorithm is the result of the optimization of decision tree algorithms and was first proposed by Breiman in 2001. Its essence is to take K samples by automatically sampling the original sample data with put-back repetitions and combining them into a new sample. Then N new features are randomly selected, and the best ones from these features are chosen as samples, and each sample is constructed as a separate decision tree. Finally, these decision trees are voted on by a random method, as shown in Equation (8). The one with the most votes is taken as the final decision result (Figure 2).

$$D(x) = \arg \max_{T} \sum_{j=1}^{K} L(d_j(x) = T) \qquad (8)$$

where $D(x)$ represents the final decision result of the random forest model, $d_j(x)$ represents the *j*th decision tree, $T$ is the predicted outcome of each decision tree, and $L()$ is the indicative function.

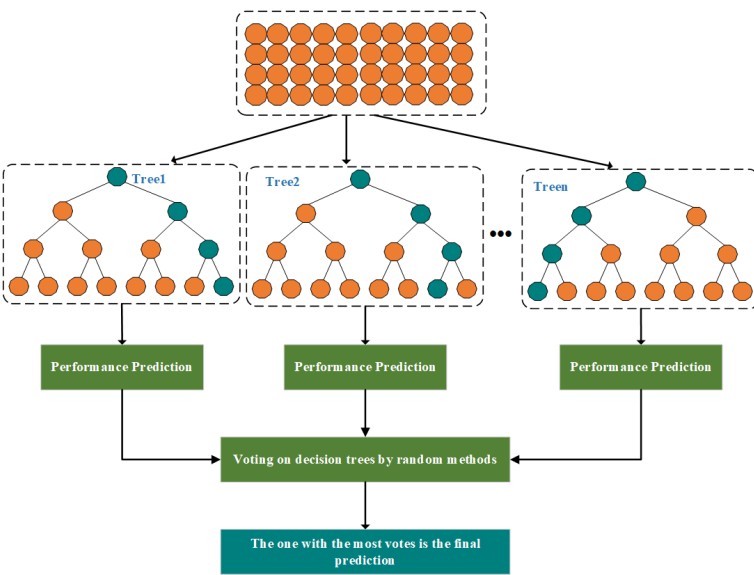

**Figure 2.** Random forest algorithm schematic.

The random forest algorithm uses the Bootstrap sampling method compared to other prediction algorithms. Hence, its subset of features is randomized, allows comparison of the importance of each feature value, generalizes well, is more tolerant of noise values, does not require specialized feature selection, and therefore, has the advantage of being used on many current datasets.

However, in the process of fluorinated-material performance prediction, this paper also found some defects of the random forest. When faced with highly unbalanced data, inaccurate classification often occurs, and at this time, if an inappropriate evaluation index is chosen, it will affect the predictiveness of the model. When the redundant features in the material dataset are complex, there are problems such as too much noise and easy overfitting after model training.

### 3.2.3. Model for Visual Analysis of Uncertainty

Uncertainty in the prediction of fluorinated materials' performances is divided into intrinsic and external uncertainty. Inherent uncertainty refers to the uncertainty of the prediction model based on the product-performance-prediction results, and external uncertainty refers to the chance factors, such as data measurement and statistics. As external uncertainty is influenced by the surrounding environment and is more contingent. This paper only discusses the intrinsic uncertainty of the performance-prediction results.

To quantify the uncertainty in the prediction of a fluorinated material's performance, this paper also explicitly constructs topological diagrams (Figure 3) of the product development's polymerization control, polymerization process, and performance-testing process.

As a result of the research, the fluorinated materials' development data and performance test data are asymptotically normally distributed. The traditional calculation of the uncertainty of normal distribution is represented by the standard deviation and variance, which are positive and negative real numbers for the test data. However, as both the fluorinated materials' development parameters and their performance prediction data are positive real numbers, this paper proposes the Latin hypercube Markov chain (LHS-MC) model for calculating the uncertainty of product prediction.

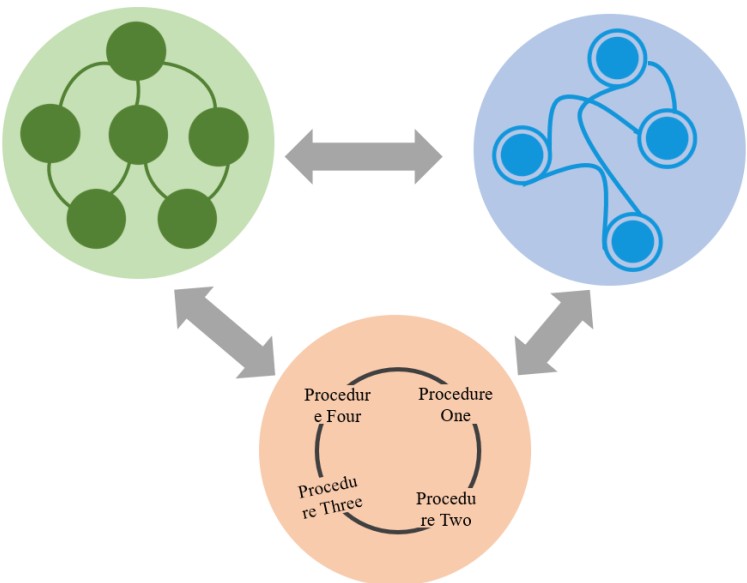

**Figure 3.** Product development process—population topology.

Given the data on the fluorinated materials' development parameters, the individual topological data were sampled using the Latin hypercube sampling method (Equation (9)).

$$x_j^{(i)} = \frac{\pi_j^{(i)} + U_j^{(i)}}{n_s}, 1 \leq j \leq n_v, 1 \leq i \leq n_s \tag{9}$$

where $i$ represents the $i$th sample point, $j$ represents the $j$th component of the $i$th sample point and $U_j$ takes the value $[0, 1]$. $n_s$ is the total number of sample points, and $\pi$ takes the value of $[0, n-1]$ as a random number.

The uncertainty of the various groups is then calculated by their variance (Equation (10)) and standard deviation (Equation (11)).

$$s^2 = \frac{\sum\limits_{i=1}^{n} (x_i - \overline{x})}{n - 1} \tag{10}$$

$$s = \sqrt{s^2} = \sqrt{\frac{\sum\limits_{i=1}^{n} (x_i - \overline{x})^2}{n - 1}} \tag{11}$$

Since the uncertainty in each population has an impact on neighboring populations, we define $p_{ij}$ as the uncertainty propagated from population i to population j, denoted by $p_i \rightarrow p_j$, and describe this uncertainty propagation as the Markov chain.

We define the uncertainty of each population topology as $E_1, E_2, \cdots, E_m$ and the transfer time between populations through the product development process as $t_1, t_2, \cdots, t_m$, using $P_{ij}^{(k)}$ to represent the probability of population $E_i$ changing $E_j$ after k steps (Equation (12)). The number of times the population $E_i$ changes $E_j$ after k steps is represented by $m_{ij}^{(k)}$, and $M_i$ means the total number of occurrences of $E_i$, so that after k steps of propagation, the transfer probability matrix is shown in Equation (13).

$$P_{ij}^{(k)} = \frac{m_{ij}^{(k)}}{M_i} \tag{12}$$

$$R^{(k)} = \begin{pmatrix} P_{11}{}^{(k)} & P_{12}{}^{(k)} & \cdots & P_{1j}{}^{(k)} \\ P_{21}{}^{(k)} & P_{22}{}^{(k)} & \cdots & P_{2j}{}^{(k)} \\ \vdots & \vdots & \vdots & \vdots \\ P_{i1}{}^{(k)} & P_{i2}{}^{(k)} & \cdots & P_{ij}{}^{(k)} \end{pmatrix} \tag{13}$$

Finally, the smooth probability distribution $\pi_i$ is calculated by Equations (14) and (15) to assist users in adjusting and optimizing the performance-prediction results by transferring the probability of uncertainty between various clusters of the product development process and improving the accuracy of decision making.

$$\sum_{j \in I} \pi_j = 1 \tag{14}$$

$$\pi_j = \sum_{k \in I} \pi_k P_{kj} \geq 0 \tag{15}$$

## 4. Visual Design

The material-performance-prediction visual analysis system was developed to assist users in evaluating the performances of materials in production and predicting the performances of materials before development. It is divided into four main modules: a control module (Figure 4a), which is used for interactive operations such as model and parameter setting and screening, and displaying data details changes; the relationship-mapping module (Figure 4b), which is used to analyze the mapping relationship between incoming material development parameters and performance; the clustering and screening module (Figure 4c), which is used to analyze the results of material performance clustering and support interactive exploration and screening by users; the performance-prediction module (Figure 4d), which is used to analyze the uncertainty factors in the material development process, combined with the machine learning model selected by the user in the control module (Figure 4a), to finally achieve the performance prediction of the material by comparing the difference between the expected performance of the material and the standard.

### 4.1. Control Module

The control module (Figure 4a) provides the user with data configuration, adjustment of material development parameters, and visualization of the performance-prediction model, designed for efficient performance prediction. It contains two essential conditions for the system's operation: the selection of material products and the selection of prediction models, with a highlighted color display for the product selection.

When the user specifies the type of material and the prediction model to be used, the parameter display unit shows the current position of the parameter in the form of a sliding axis, which supports the user in adjusting the control parameters by dragging the sliding axis in the control module. The details of each parameter adjustment are recorded as a line graph on the detail-display page. The node changes at different times are represented as nodes on the line graph using visual analysis of the time sequence. The user can drag the small circles of the parameters to adjust the control parameters and thus analyze the history of the parameter adjustment and the adjustment process. Once the parameter-adjustment operation is complete, the user can invoke the relational mapping and performance-prediction algorithm by clicking the "Run" button and visualizing the returned results in the relational mapping and performance-prediction module.

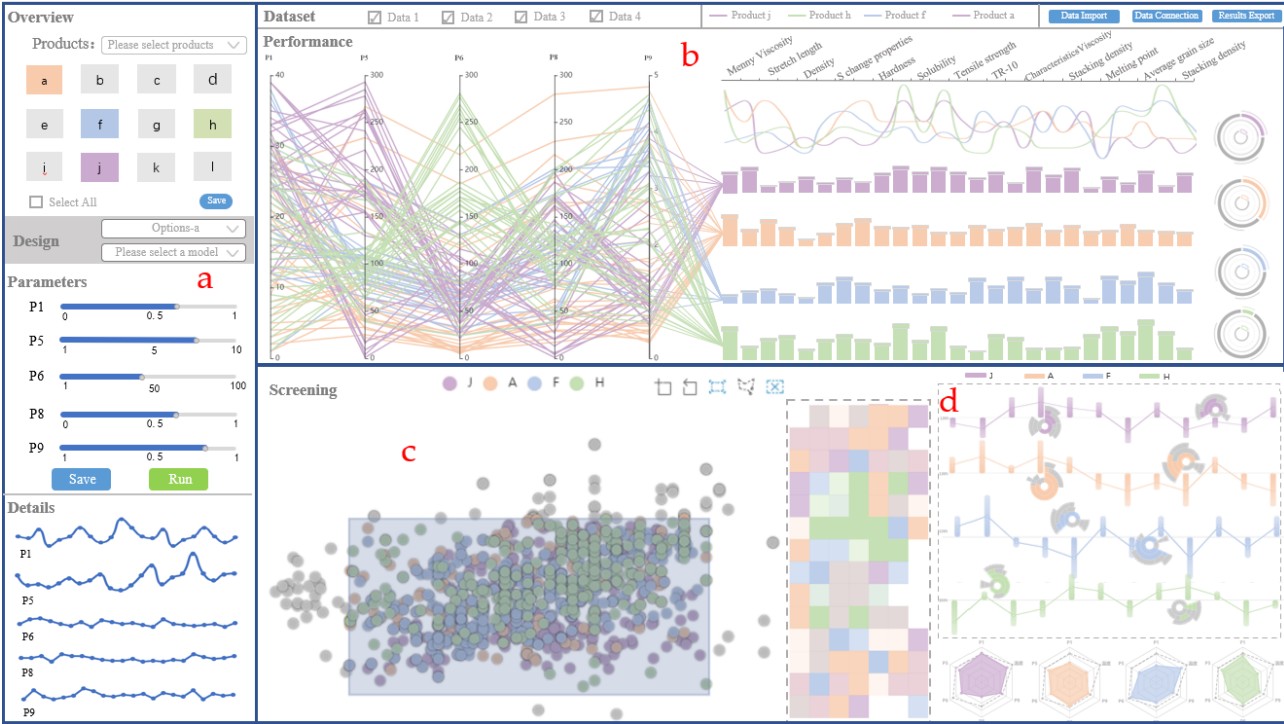

**Figure 4.** MP²-method supports users' independent interactive selection of control parameters and prediction models to realize visual analysis research of performance prediction. (**a**) Control view enables users to set and filter models and parameters. (**b**) A relational mapping system is used to analyze the mapping relationship between parameters and performance. (**c**) Clustering view provides interactive clustering analysis of user operation results. (**d**) Uncertainty visualization is introduced to visually analyze the differences between material performance and standard values for comparison.

### 4.2. Relationship-Mapping Module

The relationship-mapping module uses parallel coordinates to explore the mapping relationships between selected control parameters, uses bar charts to represent product performance, and provides visual interaction techniques to help users explore the relationships between control parameters and performance in the material development process through visual analysis of the association between product parameters and performance. Above the histogram, which represents the material's properties, a line graph is used to help the user analyze the trend of the histogram. The different colors represent the types of product selected by the user in the control module, assisting the user in analyzing them better.

- Parallel coordinates

When representing a large amount of complex data in high dimensions, the traditional parallel coordinate technique can easily lead to image clutter and cause visual confusion for users. To alleviate the user visual confusion, we adopt the visual analysis method of regional clustering and stratification to give the same color to the same types of product control parameters, and provide a lot of updates and expansions to the parallel coordinates, so that the user can select products of the same color or products within a fixed area for cluster analysis through interactive visual analysis, highlighting the advantages of cluster visual analysis and maximizing the exploration of the material. The benefits of cluster visual analysis are emphasized to maximize the investigation of the correlation between the control parameters in the process of material development.

- Results Analysis Design Solution

Once the material-development control parameters are correlated with the product performance, as shown in Figure 5, we assign a fixed centripetal angle to the traditional

rectangle and attach a selected color for each product so that it is in the inner part of the entire torus, representing the conventional value of that product's performance under the conditions. The rectangles with the same coloring and tilt are used as the outer ring of the torus, and the little gray shading used on the outermost side represents the product's performance metrics after the user explores it in a clustered, parallel-coordinate fashion. The rings in different positions form the whole ring chart (Figure 5), which facilitates the user to compare and analyze the difference between the current product performance and the standard value, and assists the user in adjusting the control parameters in the reverse direction, thereby helping the user to better explore the relationships between the control parameters and the performance data.

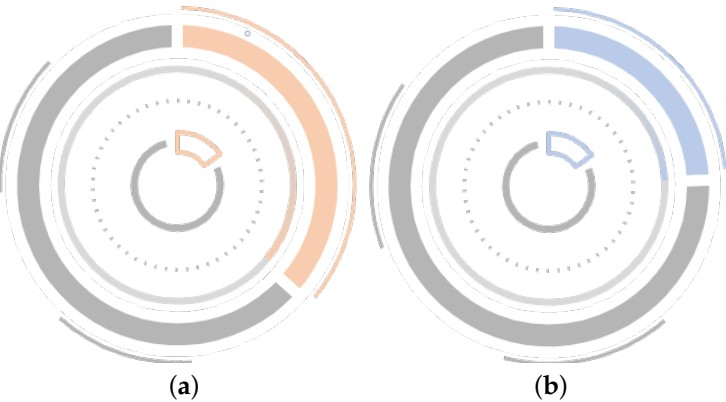

(**a**)          (**b**)

**Figure 5.** The colored area indicates the performance index of a product after analysis in a clustered parallel coordinate approach. (**a**) Correspondence between product A control parameters and product performance after screening by brushes to facilitate comparison and analysis of the difference between current performance and standard values. (**b**) Correspondence between control parameters representing product F and product performance after brush screening.

### 4.3. Performance-Prediction Module

- Clustered Scatter Plots

We visualize the results of the annulus analysis of the relationship-mapping module (Figure 6a) using a clustered scatter plot, where the color difference indicates the product type and the horizontal coordinate indicates the temporal attribute. The vertical coordinate shows the numerical magnitude of the difference between the product performance and the standard value. The points with similar difference values are clustered together using the density-peak clustering method. In addition, the clustered scatter plot supports users to hover over the points to view the values and also allows them to interactively explore the details (Figure 6b) of the differences between product performance and standard values by zooming in on the details of the boxed area for analysis through operations such as swiping.

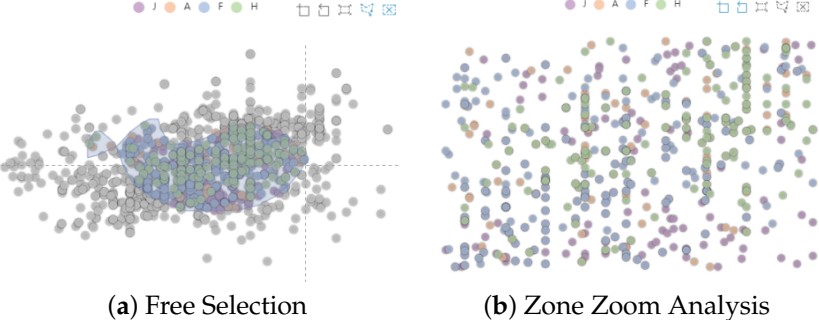

(**a**) Free Selection          (**b**) Zone Zoom Analysis

**Figure 6.** Free filtering (**a**) of aggregated areas by brushes and detailed zoomed in analysis (**b**) to interactively explore the details of how this product's performance differs from standard values.

- Uncertainty Analysis Solution

According to the results of the user-clustering scatter-plot box selection, we visualize and analyze the data of parameters with uncertainty, such as the temperature of the corresponding product, in the form of a heat map. Each color represents a product selected by the user of the control module, and we divide the transparency of the color into five stages. The darker the color, the smaller the value of the parameter representing temperature in the relational mapping module, and on the contrary, the larger the parameter value of temperature. Similar to the clustered scatter plot, the heat map also helps the user to use tools such as brushes to frame the details of some parameters to zoom in on the analysis.

- Performance Prediction Design

In the visualization of the performance-prediction results, we use a bar chart to represent the proximity between product performance and standard performance at a certain point in time, with the central horizontal coordinate representing 100% similarity, and a gradient color is used for the bars of the bar chart, which can be used to differentiate the size of the difference between a product's performance and the standard by color. Lighter colors represent more significant differences. We also use a line graph to describe the trend of product performance under time-series data and the rising sun chart rings to represent the performance indicators, where the amount of color represents the magnitude of the current value.

We use a visual analysis of the optimized radar plot (Figure 7a), where the dashed box represents the product-performance test value and the colored part in the middle refers to the predicted product-performance value, and the difference between the two is the uncertainty calculated by the uncertainty-analysis model. The difference between the two is the uncertainty calculated by the uncertainty-analysis model. The clustering optimization algorithm is also combined with a comprehensive evaluation of the material's performance by comparing the differences from the parameters selected by the user in the control module, allowing the user to adjust the parameters. In contrast, the performance-prediction system gives the user real-time feedback on the results, thereby achieving the purpose of performance prediction.

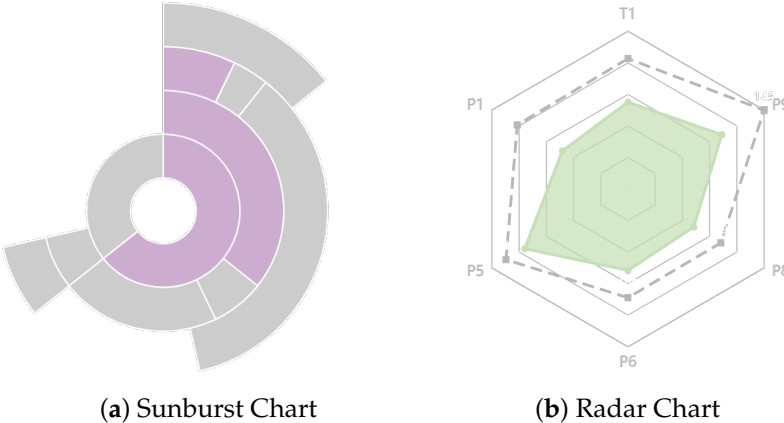

(**a**) Sunburst Chart      (**b**) Radar Chart

**Figure 7.** (**a**) The Sunburst diagram was used to visually analyze the important nodes of the variance in the gaps between the expected performance values and the standard values. (**b**) Comprehensive evaluation of the predicted performance values in the form of radar plots and analysis of the differences with the standard values of each parameter.

## 5. Case Study

In this section, we examine each of the four datasets delineated in this paper. The test datasets comprised 100 product control parameters and over 10,000 product performance data points for fluorinated materials collected over a period of 30 days from October 2020 to November 2020. To address data-security concerns, we replaced product types

and parameter names with letters. The training set was obtained from the data collected between 1 October 2020 and 25 October 2020, and the remaining datasets were used as the test set. To further validate MP$^2$-method, we conducted interviews with six domain experts who were randomly selected from our partner chemical companies, all of whom were users of the performance-prediction visual analysis system. The interviews were conducted to demonstrate the validity of MP$^2$-method. We calculated uncertainty using the LHC-MP uncertainty-analysis model that we constructed, and we analyzed the performance-prediction results using the visual analysis method of the rising sun bar chart.

The network learning rate is one of the critical parameters of the neural network model, and too large or too small a learning rate significantly impacts the prediction model's accuracy. The number of iterations is generally within the range of 100 to 500; too large or too small a number will affect the running time of the model. We determined the BP neural network's parameters displayed in Table 2 by analyzing many experimental findings.

**Table 2.** Model parameter settings table.

| Parameter Name | Parameter Value |
| --- | --- |
| Learning rate | 0.01 |
| Input dimension | 4 |
| Output dimension | 2 |
| Number of iterations | 300 |
| Minimum error of training target | 0.001 |

In the test results (Table 3) and the measurement results, MSE stands for mean square error, MAE stands for root mean square error, and $R^2$ is often referred to as the coefficient of determination, through which it is easier to see the gap between models. Usually, the value of $R^2$ is between 0 and 1, and the larger its value is, the better the accuracy of the model. By comparing and analyzing the evaluation metrics for different datasets (Figure 8), we found that when the dataset is small, Example of dataset1 (Figure 8a) vs. dataset3 (Figure 8c), the random forest prediction model is less computationally expensive and does not need to rely on a GPU to complete the training, so it has better performance. On the contrary, the BP neural network model has a strong advantage when the volume of data is large, Example of dataset2 (Figure 8b) vs. dataset4 (Figure 8d). Since there are uncertainties of variables such as temperature in the fluorinated-material development process, and the BP neural network model does not need to consider the variables, it is also proved that the application of the BP neural network model can achieve better results when the data volume is significant, especially for dataset 4. Therefore, the visual analysis method constructed in this paper, in which users independently select a suitable performance-prediction model according to their needs, can vastly improve the efficiency of material development.

During the experimental process, four products (J, A, F, and H) were selected as case studies for analysis, and the experimental approaches were categorized into three groups: Group A, Group B, and Group C. Group A consisted of material-development experts who utilized their years of experience and traditional methods, such as Excel, to predict pre-production product performance. Group B comprised those who did not utilize uncertainty models for product-performance prediction, whereas Group C utilized the LHC-MP uncertainty of visual analysis model to quantify the uncertainty in the product-performance-prediction process.

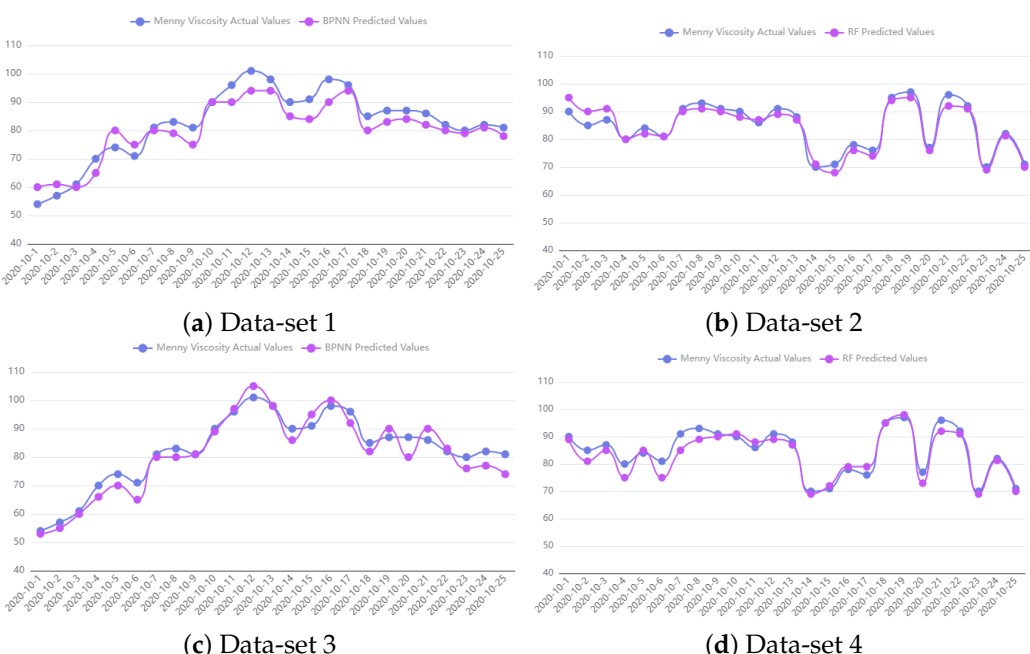

**Figure 8.** Performance-prediction analysis based on the BP neural network tested on dataset 1 (**a**) and dataset 3 (**c**); performance prediction analysis based on the random forest algorithm tested on dataset 2 (**b**) and dataset 4 (**d**).

We compare the experimental results of different groups of the four products and present bar charts to indicate the error of the experimental results of a total of 12 groups of each of the four products in relation to the standard performance of the product and expresses it as a percentage, as shown in Figure 9. Based on the analysis results, it is evident that Group C, which employed the LHC-MP uncertainty-visualization model, had the smallest error in performance prediction against the standard value for products J, A, F, and H. Additionally, Group B provided smaller results for some products and larger results for others in the development of fluorinated products compared to Group A, indicating that the use of the LHC-MP uncertainty-visualization model is indeed beneficial to the accuracy of the performance predictions of fluorinated-material products.

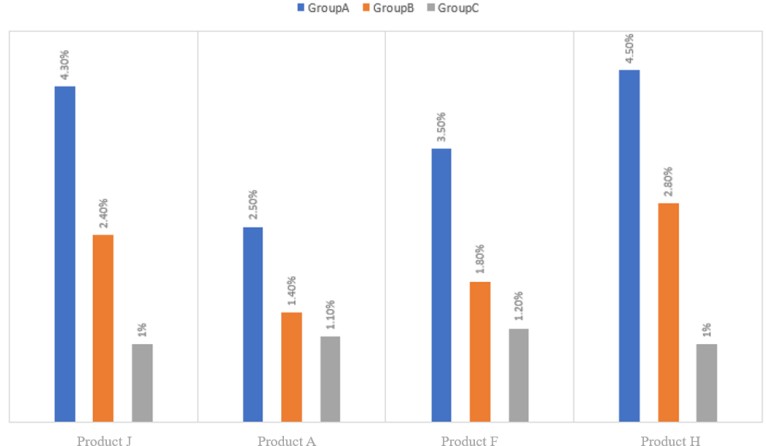

**Figure 9.** The results of the visual uncertainty analysis model are shown. The uncertainty calculation using the uncertainty analysis model LHC-MP is effective at predicting the performances of fluorinated-material products and can greatly reduce the uncertainty in the development of fluorinated-material products during the polymerization control, the polymerization process, and performance testing, and reduce the degree of error with the standard performance.

**Table 3.** Model performance data table.

| Data | Models | Train Tine (s) | MSE | MAE | $R^2$ |
|---|---|---|---|---|---|
| data-set 1 | BP Neural Network | 0.78 | 29.63 | 15.71 | 0.76 |
| data-set 2 | Random Forest | 0.36 | 12.33 | 9.11 | 0.84 |
| data-set 3 | BP Neural Network | 1.23 | 12.34 | 10.1 | 0.90 |
| data-set 4 | Random Forest | 0.96 | 30.33 | 20.61 | 0.70 |

It is noteworthy that the uncertainties in the product development and polymerization control, polymerization process, and performance testing made the performance-prediction method less accurate compared to traditional manual experience when using machine learning models alone. This factor negatively affects the final performance-prediction results of fluorinated-material products.

**6. Expert Interview**

Six domain experts from among the users of the performance-prediction visualization system were randomly invited for an interview to demonstrate the efficacy of $MP^2$-method and LHC-MP. The experts were requested to assess the accuracy of the system's visual analysis results relative to their own work experience and requirements. In addition, we sought their feedback on ways to enhance the system's design, interaction, visual analysis, and performance prediction. This feedback was recorded during the 90 min interview process.

Regarding the system's improvement and optimization, the domain experts' recommendations are as follows.

- Usefulness

The experts who were interviewed in the field unanimously agreed that our visual analysis system for material-property prediction is highly useful and can substantially reduce the number of experiments while enhancing the success rate of product development. For instance, Expert A noted that the system has fundamentally transformed the prevailing mode of material development. This innovative solution, which closely integrates computer technology with interdisciplinary materials science, enables easy processing and analysis of data. It effectively addresses their existing challenge of managing vast and complex data, while concurrently enhancing their development efficiency.

- Reliability

The experts involved in all fields appreciated the performance-prediction accuracy of $MP^2$-method. By comparing the results obtained by experts relying on empirical analysis and using the performance-prediction system, it was found that although the accuracy of the performance-prediction system needs to be improved, the problem can be solved entirely later through the improvement of the model and the optimization of the visual analysis scheme. The experts also claimed that the effective combination of a relational mapping algorithm, density peak clustering algorithm, and performance-prediction algorithm significantly improved the efficiency of performance prediction, provided fast performance screening, and reduced their mental workloads.

Although the field experts highly rated the practicality of $MP^2$-method, they also made some constructive comments. expert B pointed out that for the product development and analysis software used by chemical companies, the confidentiality measures must continue to be optimized to be foolproof. Expert D said, for new users, the system visualization design has a large amount of information, which may be temporarily unacceptable, but after a long time of use, and familiarity with the system, you will find it good use. As E experts commented, we look forward to exploring better relational mapping models and performance-prediction models to improve the accuracy of performance prediction.

## 7. Conclusions and Future Work

In this paper, we proposed $MP^2$-method, a visual analysis method for material performance prediction that supports model selection, constructs uncertainty-visualization models, and supports users' independent selection of performance-prediction models based on the optical analysis system for performance prediction. Performance prediction of the product can help users to improve their development success rates. In addition, the feasibility of a visual analysis system for performance prediction has been evaluated through four dataset case studies and interviews with domain experts. The results show that performance prediction in the material development process is valuable and reliable.

In future work, we will optimize algorithmic models such as relational mapping and clustering, refine the fluorinated-material performance-prediction research process, add user initiative to the performance metrics analysis page, and push performance data of interest to users based on their settings and preferences. In addition, we will expand the compatibility and versatility of the performance prediction visualization system so that it can be applied to other areas of performance prediction and uncertainty analysis that are affected by multiple parameters, such as traffic dynamics prediction and many other applications. We also hope to increase the performance-prediction models' accuracy to improve the system's applicability and accuracy.

**Author Contributions:** Conceptualization, Q.C. and L.Z.; methodology, Q.C.; visualization, Z.H., Q.C. and L.Z.; data curation, Z.H.; writing—review and editing, W.Z.; supervision, Y.W. All authors have read and agreed to the published version of the manuscript.

**Funding:** This study is supported in part by the National Defense Basic Research Project (JCKY2022404C001) and the Sichuan University of Science and Engineering Graduate Student Innovation Fund (y2021077).

**Institutional Review Board Statement:** Not applicable.

**Informed Consent Statement:** Not applicable.

**Data Availability Statement:** Not applicable.

**Conflicts of Interest:** The authors declare no conflict of interest.

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
