# Peer review of "A Visual Analysis Method for Predicting Material Properties Based on Uncertainty"

_applsci, doi:10.3390/app13084709_

Round 1

Reviewer 1 Report

The authors present a visual analytics system for model selection in the field of material performance prediction. 

First of all: There must have been a mistake. This article was presented to me as part of a special issue (Virtual Reality, Digital Twins and Metaverse) but its topic does not fit in at all.

In this review I would only like to name the most severe weaknesses of the paper without going into further detail:

- Related work: it seems very shallow and confusing to me. It almost seems as if the literature has been selected randomly I don't see a system here.

- Background: I do not see why the authors explain BPNN and random forest in this paper.

- Background: what is Figure 3 and what dpes this have to do with the system?

- Literature: A lot of unclear choices. For instance, how is [22] a reference for uncertainty-aware visual analysis in general?

- Expert interview: what was the method? It seems the authors just picked statements from the interviews that were positive? There is no additional material so it's hard to tell. What do we learn from the interviews? How can you overcome the complexity of the system? What do we learn for future visual analytics system design? How did they get along with the uncertainty information? etc.

- Conclusion: this almost sounds like a commercila text, saying that the system was proven to be "valuable and reliable". What part of the evaluation tells us so?

- Language: A lot of unclear sentences ("Due to the high confidentiality of fluorine material data, 3 it is not convenient for the plant to trust the data to third party professionals for processing and 4 analysis", ll. 3f) and errors "Nodel design" (l.105).

The visual analytics system seems to be useful and the design seems good but the paper does not really tell us about the system. Its purpose remains a bit nebulous to me. It would be interesting to asses whether people could make use of the uncertainty visualization or not. I recommend putting the focus on this aspect in a resubmission.

Author Response

Dear Reviewers and Editors:

Greetings!

First of all, thank you very much for taking the time out of your busy schedule to read and revise my article. Thank you for your valuable suggestions. You have made comprehensive corrections to the structure, content, research methods, and results of my thesis. It has played a very important role in improving the quality of my paper.

I have carefully studied the reviewers' comments and have carefully revised the paper according to the suggestions as follows.

  1. This paper mainly introduces neural networks and machine learning as an example to compare the effect of different models for different datasets, so that cross-industry researchers can also make performance prediction by model selection for different datasets in pre-production.
  2. Figure 3 shows the proposed Latin hypercubic-Markov chain model (LHS-MP) for calculating the uncertainty d of product prediction for product development aggregation control, aggregation process and performance testing process for constructing the topology diagram for this paper
  3. Language issues in the review comments have been revised
  4. Detailed description of the work on visualization in the article

Finally, I would like to thank you again for your guidance and for reviewing and revising my paper again. I hope that under your guidance, I can complete this excellent paper and sincerely hope that my paper will be published in your journal.

Reviewer 2 Report

The ABSTRACT section is well–presented, relevant for this subject in area of current research on material performance prediction based on machine learning. The paper present a visual analysis method for fluorine material performance prediction, MP2–Method, which uses machine learning and deep learning to explore the relationship between fluorine material control parameters and product performance to achieve product performance prediction. The general aims and objectives of the research are well defined. The paper is structured properly (INTRO, MATERIAL & METHODS, RESULTS & DISCUSSION, CONCLUSIONS, REFERENCES, etc.).

The INTRODUCTION section provide the necessary background information regarding the fluorine materials and the research on material performance prediction based on machine learning. The section is well–documented. Also, the aims and objectives of the research are well defined.

The METHODOLOGY / MATERIALS & METHODS section is relatively well described and include detailed information. The body of paper describe the important RESULTS of the research, followed by several DISCUSSIONS.

The CONCLUSION section succinctly summarize the major points of the paper, quite ambiguous, but presented concisely and to the point. But, the conclusion is intended to help the reader understand why your research should matter to them after they have finished reading the paper. A conclusion is not merely a summary of your points or a re–statement of your research problem but a synthesis of key points. I would recommend presenting the novelties of this study, the main characteristics that individualize these research.

Literature review provides comprehensive information about the current state of research. Literature sources cover the last two decades, but modern works (over the last 5 years) are mainly used.

Author Response

Dear Reviewers and Editors:

Greetings!

First of all, thank you very much for taking the time out of your busy schedule to read and revise my article. Thank you for your valuable suggestions. You have made comprehensive corrections to the structure, content, research methods, and results of my thesis. It has played a very important role in improving the quality of my paper.

I have carefully studied the reviewers' comments and have carefully revised the paper according to the suggestions as follows.

  1. Fluorine materials are compounds containing the element fluorine. In this paper, we take fluorine rubber, which is one of the most important organic fluorine materials, as an example to study.
  2. optimized the presentation of the conclusion section, which describes the novelty of this study.
  3. The parts of the related work are optimized according to the review comments and modified into the following three parts according to the research content of the article: visual analysis of association mapping, performance prediction analysis, and visual analysis of uncertainty.
  4. Optimized the quality of the references and increased the use of modern works (last 5 years).

Finally, I would like to thank you again for your guidance and for reviewing and revising my paper again. I hope that under your guidance, I can complete this excellent paper and sincerely hope that my paper will be published in your journal.

Reviewer 3 Report

Applied Sciences Journal

Title: A visual analysis method for predicting material properties based on uncertainty

Overall comments

I intend to provide the authors of this article with a holistic review that will include both major and minor comments, if any and wherever necessary. Scientific revisions fall under the major category, while typos/grammatical errors are pointed out in the minor category. Here are my comments:

·      This paper suggests an alternative pathway to the traditional machine learning approach to predict the performance data of fluorine materials, which comes from multiple sources and is complex. The prime reason for not adopting a ML approach is that they believe that 3rd party professionals cannot be trusted with the fluorine materials data since it becomes a question of national defense.

·      Instead, they introduce a visual analysis method (called MP2-method) for predicting the performance of fluorine materials by adjusting control parameters. In addition, through the proposed LHS-MC model, the uncertainty of the control parameters is improved.

·      I feel that, though these data are matters of national defense and any leak can be dangerous for the country, they can develop an in-house system for data analysis and still use internal data analysts for performing the ML techniques, in addition to the proposed techniques. Hence, I believe that completely discarding the use of ML technique for complex data is not correct since present-day ML techniques such as Random Forests decision trees, support vector machine regression, convolutional neural networks, etc. thrive on dealing with complex and heterogeneous data

·      I disagree with the statement in Line 30-31 that the predictions from ML models apply only to a particular dataset, which is not true. So many ML models have been used to predict real-time data as well as multiple datasets from various sources after building the models on different datasets. I request the authors to clarify this line

·      Please include that you are using ML and deep learning in your visual analysis method in the abstract also.

·      I suggest the use of traditional model-performance metrics such as RMSE and R2 to corroborate the uncertainty analysis of the control parameters to support your results.

·      Can this model be applied to other systems as well? Give an example in the introduction

·      Please change the title of the 2nd section to “Flow of work done in this paper” and present the work as a flowchart for better viewing and understanding, in addition to your description.

·      Does PCA or HCA not serve your purpose for analyzing multidimensional data? A clear reasoning must be provided to support your visual analysis instead of dimension reduction or factor analysis techniques.

·      In the paragraph of lines 77-87, please mention the author’s names while citing their paper as “Name et al. []”.

·      In the introduction, you say that ML techniques does not work well and suddenly you are using BPNN model in the methods for your paper. Please explain this contradiction.

·      Please mention the reference of Breiman’s work in line 166.

·      You are using BPNN and Random forest models but please specify the clear contribution of visual analysis in determining the performance of the models in your paper. It is still not clear.

·      I see great potential in your suggested method, where if all the mapping and prediction steps can be integrated into a single system enabling the user to choose the different ML techniques and see their performances, this will be a great innovation. The term ‘visual analysis’ will not appeal to data analysts because it becomes more subjective then. The authors can look to change their title by incorporating more rational terms. For example, the term “Model selection based on user experience involving ML methods..”. This can be a User Interface or a ML App in future.

·      Please include more ML models such as PLSR, i-PLSR as well as PCR for linear and collineaer data.  

·      Hence, I feel that if the authors address these comments, I can recommend this paper for publication acceptance

Author Response

Dear Reviewers and Editors:

Greetings!

First of all, thank you very much for taking the time out of your busy schedule to read and revise my article. Thank you for your valuable suggestions. You have made comprehensive corrections to the structure, content, research methods, and results of my thesis. It has played a very important role in improving the quality of my paper.

I have carefully studied the reviewers' comments and have carefully revised the paper according to the suggestions as follows.

  1. We do not abandon the use of ML techniques for complex data versus claims about specific datasets. The focus of this paper's research is to change the work that would traditionally help researchers with model selection, but rather to facilitate cross-industry researchers to more easily select models with better results for their experiments by supporting a visual analysis approach to model selection on their own.
  2. Completed reviewer comments on literature citation.
  3. This paper mainly introduces neural networks and machine learning as an example to compare the effect of different models for different datasets, so that cross-industry researchers can also make performance prediction by model selection for different datasets in pre-production.

Finally, I would like to thank you again for your guidance and for reviewing and revising my paper again. I hope that under your guidance, I can complete this excellent paper and sincerely hope that my paper will be published in your journal.

Reviewer 4 Report

The authors apply visual analysis methods to predict material properties.

There is some diligent work presented, but the material can be better structured to do justice to the authors' results.  Specifically (numbers refer to corresponding lines):

1.   16-23:  What exactly are "fluorine materials"?  Freon, Nafion, Teflon, ...?  Refs. [1] and [2] say nothing of this.

2.  24-60:  As is often the case, the authors are eager to address solutions, but it is unclear what the problem they are addressing is.  

Recommendation: 

a.  Briefly explain what fluorine materials are.

b.  Explain the problem:  What are typical examples of the nature of data and properties sought? 

c.  What is the state of the art?  I.e. what methods are currently used?

d.  What is missing/failing from the state of the art?  (A need and/or opportunity.)

I trust that the authors can address the above well, as hints of corresponding answers are already in Introduction.

3.  62-104:  The discussion is little motivating and generic.

Questions:  

a. 62-64:  Why "multidimensional data visualization, selection of prediction models, visual analysis of time-series data, and visual analysis of uncertainty."?

b. 77-84:  Where will the data come from for training these models?  

4.  Sections 3, 4, 5:  I recommend moving background material on the tools used (NNs, random forest,...) to appendices, as the methods are already well established.

Retain (a) whatever may be new, and (b) adequate guidance (in appendices and cited references) for independent reproduction of the results.

5.  344-346:  "Usually, the value of R2 is between 0 and 1, and the larger its value is, the better the accuracy of the model."
Not correct, on multiple counts:

(a) R2 is between 0 and 1 always!

(b) R2 close to one says little about the quality of the model;  it simply says that the fitted model fits the data well.  A fitted model with R2 well below 1 can still be a very good model if the low R2 is due to large noise in the data;  and a model with R2 very close to 1 says nothing about model quality, which may be excellent for interpolation but unacceptable for extrapolation.  

Various authors discuss these points in literature;  I trust the authors can cite references of their choice.

6.  Table 3 and 342-344:  Are the reported MSE/MAE/R2 for fitting or for validation or for cross-validation?
Fitting alone does necessarily create a good predictive model;  cross-validation is one of a number of approaches that helps create models with good predictive ability.

Overall:  I believe the authors could do better justice to their work by presenting it in better light.

Author Response

Dear Reviewers and Editors:

Greetings!

First of all, thank you very much for taking the time out of your busy schedule to read and revise my article. Thank you for your valuable suggestions. You have made comprehensive corrections to the structure, content, research methods, and results of my thesis. It has played a very important role in improving the quality of my paper.

I have carefully studied the reviewers' comments and have carefully revised the paper according to the suggestions as follows.

  1. Fluorine materials are compounds containing the element fluorine. In this paper, we take fluorine rubber, which is one of the most important organic fluorine materials, as an example to study.
  2. The problems pointed out by the jury on 24-60 have been improved.
  3. The parts of the related work are optimized according to the review comments and modified into the following three parts according to the research content of the article: visual analysis of association mapping, performance prediction analysis, and visual analysis of uncertainty.
  4. This paper mainly introduces neural networks and machine learning as an example to compare the effect of different models for different datasets, so that cross-industry researchers can also make performance prediction by model selection for different datasets in pre-production. The reported MSE/MAE/R2 is mainly used to validate the found problem of having different models that are better for different datasets.

Finally, I would like to thank you again for your guidance and for reviewing and revising my paper again. I hope that under your guidance, I can complete this excellent paper and sincerely hope that my paper will be published in your journal.
